# Bifunctional MoS_2_/Al_2_O_3_-Zeolite Catalysts in the Hydroprocessing of Methyl Palmitate

**DOI:** 10.3390/ijms241914863

**Published:** 2023-10-03

**Authors:** Evgeniya Vlasova, Yiheng Zhao, Irina Danilova, Pavel Aleksandrov, Ivan Shamanaev, Alexey Nuzhdin, Evgeniy Suprun, Vera Pakharukova, Dmitriy Tsaplin, Anton Maksimov, Galina Bukhtiyarova

**Affiliations:** 1Boreskov Institute of Catalysis SB RAS, 630090 Novosibirsk, Russia; 2Faculty of Natural Sciences, Novosibirsk National Research University, 630090 Novosibirsk, Russia; 3Faculty of Chemistry, Lomonosov Moscow State University, 119991 Moscow, Russia; 4Topchiev Institute of Petrochemical Synthesis RAS, 119991 Moscow, Russia

**Keywords:** biojet fuel, MoS_2_ catalyst, zeolite, Brønsted acid sites, hydrodeoxygenation, hydroisomerization, methyl palmitate

## Abstract

A series of bifunctional catalysts, MoS_2_/Al_2_O_3_ (70 wt.%), zeolite (30 wt.%) (zeolite—ZSM-5, ZSM-12, and ZSM-22), and silica aluminophosphate SAPO-11, were synthesized for hydroconversion of methyl palmitate (10 wt.% in dodecane) in a trickle-bed reactor. Mo loading was about 7 wt.%. Catalysts and supports were characterized by different physical-chemical methods (HRTEM-EDX, SEM-EDX, XRD, N_2_ physisorption, and FTIR spectroscopy). Hydroprocessing was performed at a temperature of 250–350 °C, hydrogen pressure of 3.0–5.0 MPa, liquid hourly space velocity (LHSV) of 36 h^−1^, and an H_2_/feed ratio of 600 Nm^3^/m^3^. Complete conversion of oxygen-containing compounds was achieved at 310 °C in the presence of MoS_2_/Al_2_O_3_-zeolite catalysts; the selectivity for the conversion of methyl palmitate via the ‘direct’ hydrodeoxygenation (HDO) route was over 85%. The yield of iso-alkanes gradually increases in order: MoS_2_/Al_2_O_3_ < MoS_2_/Al_2_O_3_-ZSM-12 < MoS_2_/Al_2_O_3_-ZSM-5 < MoS_2_/Al_2_O_3_-SAPO-11 < MoS_2_/Al_2_O_3_-ZSM-22. The sample MoS_2_/Al_2_O_3_-ZSM-22 demonstrated the highest yield of iso-alkanes (40%). The hydroisomerization activity of the catalysts was in good correlation with the concentration of Brønsted acid sites in the synthesized supports.

## 1. Introduction

Negative climate change caused by greenhouse gas emissions stimulates widespread expansion of the concept of “decarbonization” of the economy by reducing CO_2_ emissions. One of the approaches to solving this problem in the transport sector is the use of renewable bio-resources to obtain biofuel components that are not inferior in quality to products of petroleum origin [1,2,3]. The promising raw materials for the production of biofuel components are triglycerides of fatty acids (non-edible oils, substandard animal fats, food production waste), fatty acid esters, and free fatty acids [2,3,4,5,6]. HDO of natural lipids gives normal saturated C_15_-C_18_ hydrocarbons, which cannot be used as a drop-in paraffinic biofuel (equivalent functionally to petroleum-derived one and compatible with existing infrastructure) without further upgrading because of pour cold flow properties [2,3,4,5,6,7].

Therefore, hydroisomerization/hydrocracking of HDO products (alkanes C_15_-C_18_) is required, which gives [4,5,6,7]. Hydroprocessed ester and fatty acid (HEFA) is also known as renewable or green diesel, hydrotreated/hydrogenating vegetable oil (HVO), and hydrotreated biodiesel. Currently, processes with two catalytic stages are used in industry to obtain paraffinic biofuel components in diesel, kerosene, and gasoline ranges from lipid-based feedstocks, among them NExBTL, UOP/Eni Ecofining^TM^, UPM BioVerno, etc. [2,4,6,7]. At the first stage, hydrodeoxygenation of the feedstock is carried out over sulfide catalysts (Ni(Co)Mo/Al_2_O_3_); at the second stage (after thorough purification from sulfur-containing compounds, CO and CO_2_), hydroisomerization/hydrocracking of the obtained alkanes is carried out in the presence of catalysts based on noble metals (Pt on composite carriers, which include zeolites ZSM-22, ZSM-23, and SAPO-11).

The development of a one-stage process to obtain paraffinic biofuel components with the appropriate properties for hydroprocessing lipid-based feedstocks is an urgent task that has gotten a lot of attention in recent years [6,8]. The fatty acid methyl esters (FAME) are often used as a model compound to study the mechanism of reactions and catalytic properties of different materials in the hydroconversion of triglycerides. However, there is an opinion that it is more profitable to use the product of triglyceride transesterification, fatty acid methyl esters, which makes it possible to obtain value-added glycerol in the same process, save hydrogen, and reduce the carbon footprint [9,10,11]. In addition, the hydroconversion of FAME needs a lower reaction temperature and pressure in comparison with triglycerides [6].

It is generally accepted that the unavoidable condition for successive one-step hydroconversion of triglycerides and esters is the use of a polyfunctional catalyst that ensures the occurrence of several reactions (HDO and hydroisomerization/hydrocracking). According to a well-known concept, hydrodeoxygenation of esters first occurs to form normal alkanes with an even (via the direct hydrodeoxygenation route, with the removal of water) or odd (by decarbonylation/decarboxylation) number of carbon atoms [6,8,12]. Hydroisomerization of obtained n-alkanes proceeds via dehydrogenation to alkenes over metallic sites that are protonated by Brønsted acidic sites with carbenium ion formation and subsequent isomerization and hydrogenation [12,13]. Several articles dealing with the hydroprocessing of esters in a mixture of normal and iso-alkane have been reported over bifunctional catalysts, differing in zeolite component and metal function [6,8,14,15,16,17,18,19,20,21].

Pt/SAPO-11 bifunctional catalysts are widely studied in the hydroconversion of vegetable oils [6,8,14]. The use of Pt/Fe3SAPO-11 showed 100% conversion of FAME with 99.6% selectivity to C_15_-C_18_ alkanes and 34.8% selectivity of *iso*-C_15_-C_18_ alkanes with 34.8% in one-step hydrotreatment at 320, 4 MPa, run time 6 h in batch reactor [14]. Adding Sn increased the selectivity of *iso*-C_15_-C_18_ alkane formation over the Pt1Sn1/Fe3SAPO-11 catalyst from 34.8% to 62.7%. Despite high activity and selectivity, Pt-based catalysts are not preferred in industrial applications because of their high cost, low abundance, and sensitivity to poison. Hence, bifunctional catalysts containing transition metals (e.g., Ni, Co) have been tested in FAME hydroprocessing in recent years [15,16,17,18,19,20,21].

Ni/HZSM-5 catalysts with differing Si/Al ratios and Ni loadings were compared in the hydroprocessing of long-chain unsaturated fatty acid methyl esters [15]. Selectivity of 88.2% for C_5_-C_18_ liquid alkanes was obtained over 10 wt.% Ni/HZSM-5 (Si/Al = 25) at 280 °C, 0.8 MPa, LHSV of 4 h^−1^, and H_2_/oil molar ratio of 15, with isomerization selectivity of 27.0%. However, the conversion of FAME was only 85.1%, which decreased to 30.1% after operation for 80 h due to carbonaceous deposits. Hydroprocessing of microalgae biodiesel was performed over a 10% Ni/meso-Y zeolite catalyst [16]. A high isomerization ratio (46.4%) and selectivity to jet fuel range hydrocarbons (56.2%) were achieved, but conversion was 91.5% and Ni crystallite size was increased from 25 to 54 nm during hydroprocessing at 275 °C and 2.0 MPa. After the addition of 4% HPW to a 10% Ni/meso-Y catalyst, the production of jet fuel-range alkanes and iso-alkane selectivity increased along with an increase in strong acid density [17,18]. It was shown that Ni/meso-Y can produce 4,47% of aromatics, while Ni-based catalysts supported by Meso-ZSM-5, Meso-Hbeta, and SAPO-34 tend to produce more aromatics in the hydroconversion of microalgae oil in batch reactors at 370–410 °C for 8 h [19]. The Ni-based catalysts are prone to deactivation by coke deposition and agglomeration during the HDO process [15,16,20,21].

The comparison of Ni/SAPO-11, Co/SAPO-11, and NiCo/SAPO-11 in the hydroconversion of FAME was performed at 360–440 °C, 1.5 MPa, and a WHSV of 2.6 h^−1^ [20]. The catalyst Ni-Co/SAPO-11, containing 3% Ni and 6% Co, exhibited optimal catalytic properties, providing 100.0% conversion of FAMEs, 93.0% selectivity to C_15_-C_18_ hydrocarbons, and 36.1% of the isomerization ratio at 400 °C.

Ni/SAPO-11 and Ni_2_P/SAPO-11 catalysts were compared in hydroconversion of methyl laurate (ML) at 320–380 °C, 1.0–5.0 MPa, WHSV of 2–8 h^−1^, and H_2_/ML molar ratio of 25 [21]. Ni_2_P/SAPO-11 exhibited higher stability in comparison to Ni/SAPO-11 in HDO of ML, but both catalysts lost hydroisomerization activity. The ML conversion was close to 100% at 360 °C, 3.0 MPa, and WHSV of 2 h^−1^, while selectivity to *iso*-undecane and *iso*-dodecane decreased from 36.9% to 28.6% on Ni_2_P/SAPO-11 for 100 h. It was shown that sintering of Ni particles and formation of carbonaceous deposit were observed on spent Ni/SAPO-11, while no obvious increase of Ni_2_P particles took place, and carbonaceous deposit was a reason for the deactivation hydroisomerization activity of Ni_2_P/SAPO-11 [21].

Sulfided NiMo/SAPO-11 and NiMo/AlSBA-15 catalysts were studied in the hydroconversion of methyl stearate at 300–375 °C, 3 MPa, LHSV of 10 h^−1^, and volume H_2_/feed ratio of 600 [12]. Both NiMo catalysts provided high HDO conversion (above 99%) and isomerization activities, but NiMo/SAPO-11 exhibited a higher yield of iso-alkanes, while NiMo/AlSBA-15 catalysts additionally promoted the formation of cracked products. The authors conclude that moderate acidity and a suitable pore size of SAPO-11 provide for the formation of mono-branched isomers. MCM-41-supported sulfide Mo, CoMo, and NiMo catalysts yielded C_4_, C_5_, and C_6_ hydrocarbons and branched C_7_ and C_8_ hydrocarbons in the hydrodeoxygenation of octanoic acid at 330 °C and 1.6 MPa. The authors concluded that the higher activity of the sulfided NiMo/MCM-41 catalyst in isomerization/cracking reactions is associated with the promoting effect of MCM-41 on the acidity of the sulfided phase [22].

The efficiency of unpromoted sulfide MoS_2_ was demonstrated recently in MP hydrodeoxygenation [23], wherein alumina-supported MoS_2_ demonstrated high selectivity for the conversion of aliphatic esters through the direct HDO route without the formation of carbon oxides [24,25,26]. This property allows for the avoidance of the effect of carbon oxides on the catalyst lifetime and the additional purification of the recycle gas from CO_x_ [27]. To the best of our knowledge, the activity of MoS_2_ nanoparticles dispersed on zeolite-containing supports has not been studied yet in the hydroprocessing of aliphatic esters.

The purpose of this work is the comparative study of sulfide Mo-containing catalysts supported on granulated composite supports differing in the nature of the zeolite (Al_2_O_3_-ZSM-5, Al_2_O_3_-ZSM-12, Al_2_O_3_-SAPO-11, and Al_2_O_3_-ZSM-22) in the hydroprocessing of methyl palmitate. The MoS_2_/Al_2_O_3_-Z catalysts have been prepared with the use of organic additives, ensuring the high dispersion of MoS_2_ nanoparticles after proper sulfidation with DMDS/dodecane solution. Characterization has been performed using a wide set of different techniques to compare the MoS_2_ size/location and the acidity of the support depending on zeolite and to elucidate the possible correlations between physico-chemical and catalytic properties.

## 2. Results and Discussion

### 2.1. Catalyst Characterization

The textural properties of the synthesized supports and Mo content in the catalysts are listed in Table 1. All supports had similar textural properties: surface area—about 170 m^2^/g, pore volume—about 0.5 cm^3^/g, and pore diameter—above 20 nm. The prepared catalysts contained about 7.0 wt.% molybdenum. Such a Mo concentration was chosen to get a monolayer on the support surface (4.0 at Mo/nm^2^) [28]. Wherein it was considered that MoS_2_ localized predominantly on the alumina surface.

According to the XRD data of the synthesized supports, the alumina and corresponding zeolite diffraction lines were clearly observed (Figure 1). The supports contained a nanocrystalline alumina phase of γ-Al_2_O_3_ (PDF № 00-029-0063, the cubic cell parameter was a = 7.915 Å, the determined average size of coherently scattering domain was 7.5 nm) and corresponding crystalline phase of zeolite ZSM-5 (PDF# 00-044-0003, the determined average size of coherently scattering domain was 80 nm), ZSM-12 (PDF# 00-086-2634, a = 24.863 Å, b = 5.012 Å, c = 24.372 Å β = 107.7⁰, the determined average size of coherently scattering domain was 45 nm), ZSM-22 (PDF# 00-038-0197, a = 13.83 Å, b = 17.41 Å, c = 5.042 Å, the determined average size of coherently scattering domain was 45 nm), and silicoaluminophosphate SAPO-11 Al_2_Si_0.35_P_1.74_ O_8.05_ (PDF# 00-047-0614, the determined average size of coherently scattering domain was 70 nm).

SEM pictures of zeolite materials and the final supports of Al_2_O_3_-zeolite are shown in Figure 2. Zeolite fragments presented on SEM images of composites support evidence of the preservation of zeolite structure in synthesized supports of Al_2_O_3_-zeolite. Moreover, EDX mapping of Al_2_O_3_-zeolite supports demonstrates a uniform distribution of zeolite in support granules. Zeolites in the synthesized supports of Al_2_O_3_-zeolite display different average particle sizes (930, 1010, 300, and 220 nm for ZSM-5, ZSM-12, SAPO-11, and ZSM-22), and their histograms of particle size distribution are given in Figure 3 (the scale was chosen so that the difference was visually seen).

The hydroxyl cover of Al_2_O_3_ and Al_2_O_3_-zeolite supports was studied by FTIR spectroscopy (Figure 4). The spectrum of pure alumina shows the vibration bands at ca. 3790, 3775, 3727, 3700–3685, and 3660 cm^−1^, which are typical for the FTIR spectrum of surface OH groups of γ-Al_2_O_3_ [29] and characterize the different types of terminal Al-OH and bridged Al−O(H)-Al groups. The spectra of alumina-zeolite composites present two groups of signals in the region of O-H stretching vibrations assigned to the hydroxyl groups of the zeolites and the alumina binder. The intensity of bands at 3790, 3770, 3727, and 3685–3700 cm^−1^ in the spectra of composites (except for the Al_2_O_3_-SAPO-11) is proportional to the binder content. In the spectrum of the Al_2_O_3_-SAPO-11 sample, a decrease in the intensity of the bands of binder hydroxyl is observed, possibly caused by the interaction of phosphate ions from SAPO-11 both with Al-OH and Al-O(H)-Al groups of alumina. The signal at 3676 cm^−1^ in the spectrum of this composite characterizes P-OH groups either in the structure of the PO_4_ tetrahedron at the external surface of silica aluminophosphates [30] or at the surface of PO_4_-doped alumina [31]. The framework Si-O(H)-Al groups of SAPO-11, corresponding to strong Brønsted acid sites (BAS), appear at 3628 cm^−1^ for Al_2_O_3_-SAPO-11 composite in accordance with [30] at 3602 cm^−1^ for Al_2_O_3_-ZSM-22 [32], at 3612 cm^−1^ for Al_2_O_3_-ZSM-5 [33] and at 3612 and 3575 cm^−1^ for Al_2_O_3_-ZSM-12 composites [34]. The intensity of the bands of bridged hydroxyls in zeolite channels for Al_2_O_3_-ZSM-5 and Al_2_O_3_-ZSM-22 composites is significantly higher than for Al_2_O_3_-ZSM-12 and Al_2_O_3_-SAPO-11 ones. The bands of hydroxyl groups attached to partially extra-framework Al-OH species of zeolites overlap with the peaks of bridged Al−O(H)-Al groups of Al_2_O_3_. The bands at 3745 and 3738–3740 cm^−1^ in the spectra of Al_2_O_3_-zeolite extrudates are assigned to terminal silanols and defect Si-OH groups located in close vicinity to the lattice imperfection or Lewis acid sites at the external surfaces of zeolite crystals [35], respectively.

The acid properties of the Al_2_O_3_-zeolite supports were studied by FTIR spectroscopy with progressive CO adsorption at liquid nitrogen temperature. Adsorption of CO on pure Al_2_O_3_ at low pressures (spectra not shown) leads to the appearance of bands at 2241, 2235, 2220–2218, and 2208–2206 cm^−1^, assigned to the coordinately bonded CO complexes with strong and moderate Lewis acid sites (LAS) [36]. An increase in CO pressure leads to the appearance of a band at about 2200 cm^−1^, red shifted to 2184–2186 cm^−1^ at increasing coverage, which is attributed to the CO complex with weak LAS of alumina. The signals at 2164 and 2158–2156 cm^−1^ indicate CO complexes with different types of Al−OH groups. The spectra of CO adsorbed on Al_2_O_3_-zeolite supports present bands related to CO adsorption both on pure alumina and on zeolites (Figure 5). The bands at 2225–2230 cm^−1^, which are attributed to the complexes of CO with strong LAS of zeolites, overlap with the same bands of CO complex with Al_2_O_3_ species. The concentration of strong and moderate LAS in Al_2_O_3_-zeolite composites varies insignificantly (except Al_2_O_3_-SAPO-11); the amount of weak LAS is the same and proportional to the alumina content in the composites. An increase in the concentration of moderate LAS with the band at 2206 cm^−1^, apparently related to Al^3+^ species modified by PO_4_^2-^ groups [31], is observed for the Al_2_O_3_-SAPO-11 support (Appendix A).

The spectra of Al_2_O_3_-zeolite samples demonstrate additional signals at 2178–2170 and 2137–2138 cm^−1^ at the CO stretching region compared to the spectra of pure alumina. The first group of bands refers to CO complexes with BAS; the second peak characterizes physically or liquid-like adsorbed CO molecules in zeolite channels [33]. The spectra of Al_2_O_3_-ZSM-5 and Al_2_O_3_-ZSM-22 supports exhibit one signal for CO complexes with strong BAS at 2176 cm^−1^, which corresponds to the spectra of pure zeolites [37,38,39]. The spectra of Al_2_O_3_-SAPO-11 composite present one band for CO complexes with BAS at 2173 cm^−1^, red shifted to 2170 cm^−1^ at increasing coverage, which corresponds to moderate BAS in accordance with the value of CO-induced blue shift relative to the CO gas phase (Δν_CO_ = 30–27 cm^−1^). The band at 2178 cm^−1^ in the spectra of Al_2_O_3_-ZSM-12 support is assigned to the CO complex with strong BAS, while the signal at 2171 cm^−1^ belongs to the CO complex with moderate BAS. The two types of BASs, framework Si-O(H)-Al groups and extra-framework Al-OH groups, are also observed in the spectra of the original ZSM-12 zeolite [34]. The CO complex with Na^+^ impurities in the ZSM-12 zeolite additionally increases the intensity of the band at 2171–2170 cm^−1^ in the case of the Al_2_O_3_-ZSM-12 composite.

The weakly basic CO molecule is known to be a good probe molecule for testing the strength of BAS in zeolites and related materials [40]. During low-temperature CO adsorption on Al_2_O_3_-zeolite samples, the bands of acidic OH groups fully disappeared, and a new band appeared (Figure 6). The red frequency shift of OH stretching vibration at hydrogen bonding with carbon monoxide is traditionally used to estimate the acidity of hydroxyl groups. A new positive peak at about 3285 and 3300 cm^−1^ appears in the spectra of Al_2_O_3_-ZSM-22 and Al_2_O_3_-ZSM-5 samples, respectively, at low CO pressure. The value of the red frequency shift of the bands from the framework Si-O(H)-Al groups at hydrogen bonding with CO (Δν_OH…CO_) is 317–320 cm^−1^ and similar to the magnitude for initial zeolites [37,38,39]. The corresponding blue frequency shift of the CO stretching bands for these composites is also the same (Δν_CO_ = 33 cm^−1^), which indicates a similar high acidity of the bridged hydroxyls. Quantitative data on BAS concentration and acid strength are given in Table 2. A large concentration of strong BAS for the Al_2_O_3_-ZSM-22 composite is obviously associated with a lower Si/Al ratio in the structure of the zeolite used. The shoulder at about 3400 cm^−1^ in the spectra of Al_2_O_3_-ZSM-5 and Al_2_O_3_-ZSM-22 supports changing in synchrony with the band at 3285–3300 cm^−1^ due to Fermi resonance [41]. Other positive bands appear at 3470–3480 cm^−1^ in the spectra of these composites and are related to hydrogen-bonded CO complexes with defect silanol groups (Si-O(H)…Al^3+^). The shift values (Δν_OH…CO_ = 270–260 cm^−1^) are slightly lower than the magnitude typical for bridged Si-O(H)-Al groups in the zeolite channel and correspond to moderately strong BAS. The concentration of these sites is negligible.

After CO adsorption on the Al_2_O_3_-ZSM-12 sample, the appearance of the new positive signal at 3285 cm^−1^ is observed. The values of the red frequency shift in the hydroxyl region (Δν_OH…CO_ = 333 cm^−1^) and the corresponding blue frequency shift in the carbonyl region (Δν_CO_ = 35 cm^−1^) in the spectra of this support are assigned to BAS with enhanced acidity that bridged hydroxyl in the zeolite channel for Al_2_O_3_-ZSM-22 and Al_2_O_3_-ZSM-5 composites. The magnitude of shifts is slightly higher than in pure zeolite [34]. The low concentration of strong BAS for the Al_2_O_3_-ZSM-12 composite compared to the Al_2_O_3_-ZSM-5 composite may be due to their partial exchange with Na^+^ impurities. The other band in the O–H stretching region after CO adsorption on the Al_2_O_3_-ZSM-12 composite is detected at ~3460–3490 cm^−1^ and is related to perturbation both of the extra-framework Al-OH groups of zeolites with the band at about 3670–3675 cm^−1^ and the defect silanols with the band at 3738–3740 cm^−1^. Apparently, ZSM-12 zeolite is partially dealuminated. According to the values of the red frequency shift, the extra-framework Al-OH groups in this zeolite are BAS with medium strength. Progressive CO adsorption on the Al_2_O_3_-SAPO-11 composite leads to the appearance of a strong positive band at 3378 cm^−1^, with the shoulder at 3470 cm^−1^, due to perturbation of bridged Si-O(H)-Al groups in the zeolite channels. The shift value (Δν_OH…CO_ = 258 cm^−1^) is significantly lower than the magnitude typical for bridged Si-O(H)-Al groups in the pure SAPO-11 channel (Δν_OH…CO_ = 310 cm^−1^) [42] and corresponds to moderately strong BAS. The change in the acidity of bridged Si-O(H)-Al groups in the zeolite channels can be probably caused by disruption of the SAPO-11 structure by the partial removal of phosphate groups during the molding of extrudates. The red shift of P-OH groups in zeolites after CO adsorption (Δν_OH…CO_ = 202 ÷ 198 cm^−1^) corresponds to somewhat weaker Brønsted acid sites.

Thus, the strength of framework BAS (bridged Si-O(H)-Al groups in zeolite channels) decreases in the series of Al_2_O_3_-zeolite supports as Al_2_O_3_-ZSM-12 > Al_2_O_3_-ZSM-22 ~ Al_2_O_3_-ZSM-5 >> Al_2_O_3_-SAPO-11, while the concentration of strong and moderate BAS of zeolites decreases in the following order: Al_2_O_3_-ZSM-22 > Al_2_O_3_-SAPO-11 > Al_2_O_3_-ZSM-5 >> Al_2_O_3_-ZSM-12.

In FTIR difference spectra during adsorption of CO on pure alumina, there are no bands in the region of 3200–3500 cm^−1^ [43], which are characteristic of CO complexes with BAS of zeolites. The terminal Al-OH groups of alumina are traditionally assigned basic properties, while the bridging hydroxyls have been shown to have weak acidic properties (Δν_OH…CO_ = 130 ÷ 100 cm^−1^). The formation of CO complexes with the Al-O(H)-Al groups of alumina during CO adsorption on Al_2_O_3_-zeolite supports occurs after saturation of the zeolite BAS (Appendix A).

According to HRTEM images (Figure 7), a dispersed sulfide phase is presented on the surfaces of the sulfided catalysts, which is visualized as a black line (edges of MoS_2_ particles). The average size of nanoparticles varied from 4 to 6 nm; the stacking number was 1.5–1.7 for all catalysts. It should be noted that MoS_2_ nanoparticles were predominantly located on the alumina surface, and only a single species was presented on the surface of zeolite. This statement is illustrated for MoS_2_/Al_2_O_3_-SAPO-11 and MoS_2_/Al_2_O_3_-ZSM-22 catalysts in Figure 7. EDX mapping confirms this statement: sulfide species (Figure 8, green color) are more prevalent on alumina surfaces in comparison with zeolite surfaces (Figure 8, red color), where sulfide particles are far less prevalent.

### 2.2. The Effect of Zeolite Type on Hydrodeoxygenation of Methyl Palmitate

The conversion of fatty acid esters can follow two routes: ‘direct’ hydrodeoxygenation (‘direct’ HDO) and hydrodecarboxylation/hydrodecarbonylation (DeCOx). In the presence of a MoS_2_ catalyst, the conversion of fatty acid esters proceeded mainly via a ‘direct’ hydrodeoxygenation pathway to form hexadecane (C_16_H_34_) and water, with the formation of carbon oxides only in trace amounts [25,44,45].

Hydrodeoxygenation (HDO) of methyl palmitate (MP) was performed at a temperature range of 250–350 °C, at H_2_ pressure of 3.0 MPa, H_2_/feed ratio of 600 Nm^3^/m^3^, and LHSV of 36 h^−1^. Methyl palmitate conversion is increased with the temperature rising from 250 to 310 °C (Figure 9). Hexadecanol, hexadecanal, palmitic acid, palmityl palmitate, and methyl hexadecyl were detected as oxygen intermediate products over MoS_2_/Al_2_O_3_-zeolite catalysts in MP hydrodeoxygenation, in consistency with the previous results [23,24,46]. At a temperature range of 250–290 °C, normal and unsaturated C_15_-C_16_ alkanes were also observed.

Figure 9 shows that the addition of zeolite to alumina has a slight influence on MP conversion. Conversions of all-oxygen-containing compounds, including both intermediates and methyl palmitate, were calculated using the contents of oxygen in the reaction mixture before and after the reaction by means of elemental analysis (Equations (1) and (2)), and the results are presented in Figure 10. According to these results, the addition of zeolite to the support leads to an increase in the conversion of oxygen-containing compounds. Taking into account that the conversion of methyl palmitate weakly depends on the composition of the carrier, we can conclude that the addition of zeolite leads to an acceleration of the HDO reactions of intermediate oxygen-containing compounds [12].

Complete MP and oxygen conversion were achieved at 310 °C in the presence of all catalysts (Figure 9 and Figure 10). Normal and iso-alkanes (C_15_ and C_16_) were detected under conditions where complete oxygen conversion was achieved (at temperatures above 310 °C). Cracked products were detected in negligible amounts over MoS_2_/Al_2_O_3_-SAPO-11 and MoS_2_/Al_2_O_3_-ZSM-22 catalysts: 2 and 4% at 350 °C, respectively. The maximum yield of cracked products was observed for MoS_2_/Al_2_O_3_-ZSM-5 (18%) and MoS_2_/Al_2_O_3_-ZSM-12 (12%) catalysts.

The selectivity for the conversion of methyl palmitate via the ‘direct’ hydrodeoxygenation route in the presence of MoS_2_/Al_2_O_3_-zeolite catalysts was over 85% (Figure 11). Temperature increase leads to a decrease in the selectivity of the C_16_H_34_ formation via the ‘direct’ HDO route over all catalysts due to occurring DeCOx reactions (Figure 11) [47]. It can be seen that the addition of zeolite to alumina resulted in an enhancement of the DeCOx route in the hydroprocessing of MP over sulfide catalysts (Figure 11). The lowest HDO selectivity was observed over the MoS_2_/Al_2_O_3_-ZSM-22 catalyst. It can be explained by the highest concentration of strong BAS on the Al_2_O_3_-ZSM-22 support surface (Table 2) that could favor hydrodecarboxylation/hydrodecarbonylation reactions of methyl palmitate. Methane and negligible amounts of carbon monoxide were detected in the gas phase.

The catalyst stability in the hydrodeoxygenation of methyl palmitate was checked after 40 h at a temperature of 290 °C. Oxygen conversion was changed slightly: from 81.5 to 80.0% for MoS_2_/Al_2_O_3_-ZSM-5, from 81.2 to 77.0% for MoS_2_/Al_2_O_3_-ZSM-12, from 85.5 to 82.9 for MoS_2_/Al_2_O_3_-ZSM-22, and from 72.7 to 68.0% for MoS_2_/Al_2_O_3_-SAPO-11. Thus, the change in catalyst activity during the experiment can be neglected.

### 2.3. The Effect of Zeolite Type on Hydroisomerization of Methyl Palmitate

The isomerization process over MoS_2_/Al_2_O_3_-zeolite catalysts was studied under the conditions of complete conversion of oxygenates, i.e., temperature above 310 °C and pressure between 3.0 and 5.0 MPa. The catalytic activity of the sulfide samples during the hydroisomerization of methyl palmitate was compared by the yield of isomeric C_16_H_34_ and C_15_H_32_ alkanes in the reaction products.

According to the obtained results, the yield of iso-alkanes gradually increases in the following order: MoS_2_/Al_2_O_3_ < MoS_2_/Al_2_O_3_-ZSM-12 < MoS_2_/Al_2_O_3_-ZSM-5 < MoS_2_/Al_2_O_3_-SAPO-11< MoS_2_/Al_2_O_3_-ZSM-22: yield of iso-alkanes did not exceed 5% over MoS_2_/Al_2_O_3_, 13.5% and 7.4% for MoS_2_/Al_2_O_3_-ZSM-5 and MoS_2_/Al_2_O_3_-ZSM-12 samples, accordingly at 310 °C, 3.0 MPa, 600 Nm^3^/m^3^, 36 h^−1^. In the presence of the MoS_2_/Al_2_O_3_-SAPO-11 catalyst, the yield of iso-alkanes increases to 24%. The most active catalyst in MP hydroisomerization was MoS_2_/Al_2_O_3_-ZSM-22, with a yield of isomerized C_16_H_34_ and C_15_H_32_ alkanes of 40% (Figure 12). The observed sequence coincides with the increase in the BAS concentration order of zeolite-containing supports: Al_2_O_3_-ZSM-12 << Al_2_O_3_-ZSM-5 < Al_2_O_3_-SAPO-11 Al_2_O_3_-ZSM-22. The hydroisomerization activity of sulfide catalysts is proportional to the number of BAS [12,48].

It was observed that catalytic properties depend not only on the acidity of samples but also on the pore structure and framework topology of zeolites in the catalyst’s composition [49]. The MP molecule has a length of 22 Å and a width of 2.2 Å (Figure 13). According to the literature, data to isomerize MP molecules should be available to diffuse into pores and channels of zeolite [12]. Catalytic experiments showed that catalysts prepared with ZSM-22 and SAPO-11 demonstrated better performance in the hydroisomerization of methyl palmitate. It is correlated with the BAS concentration of synthesized zeolite-containing supports. Moreover, the better performance of ZSM-22- and SAPO-11-containing catalysts could probably be explained by the smaller average crystallite size of zeolite in comparison with catalysts prepared with ZSM-5 and ZSM-12 (Figure 3). We can propose that zeolite with a smaller crystallite size gives a more uniform (homogeneous) distribution in the support, which in turn provides closer proximity between zeolite and sulfide entities. There is no consensus in the literature about the influence of zeolite particle size on the efficiency of zeolite-containing catalysts in hydroprocessing [50,51,52,53,54]. Acidity is probably a more significant factor than the pore structure and framework topology of zeolites.

A temperature increase from 310 to 350 °C resulted in a decrease in iso-alkane yield over all MoS_2_/Al_2_O_3_-zeolite catalysts: from 40% to 26% over MoS_2_/Al_2_O_3_-ZSM-22; from 24% to 14% in the presence of MoS_2_/Al_2_O_3_-SAPO-11 catalyst (Figure 12). Catalytic experiments showed a decrease in iso-alkane yield with a temperature rise accompanied by an increase in normal C_16_ and C_15_ alkanes, while the content of cracked products changed slightly under the reaction conditions. Currently, we do not have a reasonable explanation for the observed dependence; a thorough study of the mechanism of ester and HDO intermediate transformation may help to elucidate this issue in the future.

In addition, the effect of pressure (3.0 and 5.0 MPa) on the MP hydroisomerization over catalysts containing ZSM-22 zeolite and SAPO-11 was also investigated. The reaction was carried out at a temperature of 350 °C, an LHSV of 36 h^−1^, and a H_2_/feed ratio of 600 Nm^3^/m^3^. A pressure increase from 3.0 to 5.0 promoted MP conversion via the ‘direct’ HDO route: HDO selectivity increased from 88.4 to 90.7% over MoS_2_/Al_2_O_3_-SAPO-11 and from 85.4 to 88.9% over MoS_2_/Al_2_O_3_-ZSM-22 catalyst, in agreement with previous results [15,25,47]. The yield of iso-alkanes decreases with pressure increase from 26% to 14.5% over MoS_2_/Al_2_O_3_-ZSM-22 catalyst and from 15% to 10% over MoS_2_/Al_2_O_3_-SAPO-11 sample (Figure 14). The reason for this could be the acceleration of hydrogenation of olefins, which, according to the generally accepted mechanism, are intermediate products in hydroisomerization and hydrocracking reactions [12].

In the literature, pressure and temperature increases resulted in an increase in the iso-alkane yield [57]. The authors performed MP hydrotreating at a high temperature of 350–410 °C and a pressure of 6.0–12.0 MPa over sulfided MoO_3_/ZrPO_x_ in a batch reactor. High temperatures activated the stable alkanes, and the yield of iso-alkanes increased. Our catalytic tests were performed in a lower temperature and pressure range. A decrease in the activity of sulfide catalysts in the hydroisomerization of methyl palmitate was observed with increasing pressure and temperature, which is related to the reaction mechanism. The conversion of methyl palmitate over sulfide catalysts is quite complex, including hydrodeoxygenation and hydroisomerization reactions. Presumably, alkane isomers are formed not from the final product of hydrodeoxygenation (n-hexadecane), but from intermediate products of methyl palmitate conversion (alcohol and olefins).

## 3. Materials and Methods

### 3.1. Support Preparation

Four high-silica zeolite powders with different framework types were used to prepare catalysts (Table 3). All samples (except ZSM-12) were purchased from Zeolyst Corp.

The synthesis of zeolite ZSM-12 was carried out using the following reagents: a colloidal solution of silicon dioxide LUDOX HS-40 (40 wt.%, Sigma-Aldrich), aluminum sulfate octadecahydrate (Al_2_(SO_4_)_3_·18H_2_O, Sigma-Aldrich, 99%), methyltriethylammonium chloride ([CH_3_N(C_2_H_5_)_3_]Cl, Sigma-Aldrich, 97%, abbreviated [MTEA]Cl), sodium hydroxide (NaOH, Komponent-Reaktiv, 98%), and ammonium nitrate (NH_4_NO_3_, Khimmed, 98%). Solution A, consisting of 12.6 g of distilled water, 0.4 g of Al_2_(SO_4_)_3_·18H_2_O, 1 g of NaOH, and 3.3 g [MTEA]Cl used as a template, was stirred until all of the components became completely dissolved. Solution B, consisting of 25.2 g of a 40% (wt.) colloidal solution of silicon dioxide of the brand LUDOX HS-40 and 10.1 g of distilled water, was stirred until the reaction mixture was homogeneous. Solution A was dropped into solution B and stirred gently. The gel was poured into a Teflon liner and placed into the autoclave, which was heated at 155 °C for 120 h. The product was filtered off, washed with distilled water, dried at 110 °C for 12 h, and calcined at 550 °C for 10 h (heating rate 1 deg∙min^−1^). To obtain the H-form of zeolite, the synthesized material was treated 3 times with a 1 M aqueous solution of NH_4_NO_3_ at 80 °C for 17 h. The solid product was filtered off, washed with distilled water, dried at 110 °C for 12 h, and calcined at 550 °C for 8 h (heating rate 1 deg∙min^−1^).

Alumina support was prepared by HNO_3_ peptization of pseudoboehmite (Disperal 20, Sasol Germany GmbH, Hamburg, Germany). Zeolite-containing granular supports were prepared by mixing pseudoboehmite (Disperal 20, Sasol Germany GmbH, Hamburg, Germany) and zeolite powders, followed by peptization with nitric acid, and then piston extrusion through a trefoil-shaped die. After extruding, support granules were dried at 110 °C for 12 h and then calcined at 550 °C in air flow for 6 h. Zeolite content was 30 wt.% in all calcined composite supports. Synthesized supports were denoted as Al_2_O_3_-ZSM-5, Al_2_O_3_-ZSM-12, Al_2_O_3_-ZSM-22, and Al_2_O_3_-SAPO-11.

### 3.2. Catalyst Preparation

Mo catalysts were prepared by incipient wetness impregnation of synthesized alumina and zeolite-containing extrudates by aqua solution containing ammonium heptamolybdate ((NH_4_)_6_Mo_7_O_24_ 4H_2_O from Vekton, Saint Petersburg, Russia) and citric acid monohydrate (C_6_H_8_O_7_⋅H_2_O from Vekton, Saint Petersburg, Russia). Mo content was about 7.0 wt.% after calcination of the catalysts at 550 °C for 4 h.

### 3.3. Support and Catalyst Characterization

The textural properties of the synthesized supports were determined using nitrogen physisorption at 77 K with an Autosorb-6B-Kr instrument (“Quantachrome Instruments”, New York, NY, USA).

The elemental analysis was performed using inductively coupled plasma atomic emission spectroscopy (ICPAES) on Optima 4300 DV (“Perkin Elmer”, Norwalk, CT, USA). The Mo content was determined after calcination of the catalysts at 550 °C for 4 h.

X-ray powder diffraction (XRD) patterns of supports and catalysts were obtained with an instrument STOE STADI MP (“STOE”, Darmstadt, Germany) with a detector MYTHEN2 1K using MoKα radiation (wavelength λ = 0.7093Å). The measurements were carried out in a range of 2θ from 2 to 40°, with a scanning step of 0.015°.

The acidity of Al_2_O_3_-zeolite supports and pure Al_2_O_3_ was characterized by FTIR spectroscopy of adsorbed carbon monoxide. FTIR spectra were recorded on a Shimadzu FTIR-8300 spectrometer (Shimadzu, Tokyo, Japan) within the spectral range of 700–6000 cm^−1^, resolution of 4 cm^−1^, and 300 scans for signal accumulation. The powder samples were pressed into thin, self-supporting wafers of 0.010–0.012 g × cm^−2^ density and pretreated in a home-made IR cell at 500 °C for 2 h under a dynamic vacuum of less than 10^−3^ Pa. In the presented spectra, the absorbance was normalized to sample wafer density. CO was introduced at liquid nitrogen temperature in doses from a low pressure of 0.1 mbar up to an equilibrium pressure of 10 mbar. The concentration of Brønsted acid sites (BAS) was determined from the integral intensity of the bands assigned to hydrogen-bonded complexes of CO molecules with the OH groups using the following molar integral absorption coefficient values: *A*_0_ = 54 cm/μmol for the complexes with ν_OH...CO_ ~ 3280–3380 cm^−1^ and *A*_0_ = 27 cm/μmol for the complexes with ν_OH...CO_ 3500 cm^−1^ [61].

The morphology of supports was studied using a Hitachi Regulus SU8230 FESEM scanning electron microscope (Hitachi, Tokyo, Japan) with an accelerating voltage of 2 and 5 kV in the modes of secondary (SE) and backscattered (BSE) electrons using an upper (U) detector, which makes it possible to obtain microscopic images in phase and topographic contrasts. The study of the chemical composition was also carried out on a Hitachi Regulus SU8230 FESEM scanning electron microscope (Hitachi, Tokyo, Japan) with an accelerating voltage of 20 kV. The device is equipped with an AztecLive (Oxford Instruments, Oxford, UK) energy-dispersive X-ray characteristic spectrometer (EDX) with a semiconductor Si detector with an energy resolution of 128 eV.

The morphology of the sulfide phase of the catalysts after hydroprocessing was studied by high-resolution transmission electron microscopy (HRTEM) using a ThemisZ electron microscope (“Thermo Fisher Scientific”, Waltham, MA, USA) with an accelerating voltage of 200 kV and a limiting resolution of 0.07 nm. Images were recorded using a Ceta 16 CCD array (“Thermo Fisher Scientific”, Waltham, MA, USA). The instrument is equipped with a SuperX (“Thermo Fisher Scientific”, Waltham, MA, USA) energy-dispersive characteristic X-ray spectrometer (EDX) with a semiconductor Si detector with an energy resolution of 128 eV. To obtain statistical information, the structural parameters of ca. 500 particles were measured.

### 3.4. Catalytic Experiments

The catalytic experiments were performed using an experimental setup with a trickle-bed reactor with an inner diameter of 12 mm and a length of 370 mm. In each experiment, 0.5 mL of catalyst (0.25–0.50 mm size fraction) was diluted with inert material, carborundum (0.1–0.25 mm size fraction), in a 1:8 volume ratio. Prior to the catalytic experiments, the catalysts were activated by in-situ sulfidation with dimethyl disulfide in dodecane (0.6 wt.% sulfur) at H_2_ pressure—3.5 MPa, H_2_/feed ratio—300 Nm^3^/m^3^, and LHSV—20 h^−1^. Sulfidation was performed at a temperature of 340 °C for 4 h with a heating rate of 25 °C/h.

Hydroprocessing of methyl palmitate was carried out at a temperature range of 250–350 °C, H_2_ pressure of 3.0 and 5.0 MPa, H_2_/feed ratio of 600 Nm^3^/m^3^, and LHSV of 36 h^−1^. The feed was 10 wt.% of methyl palmitate (1.17 wt.% O) with dimethyl disulfide (0.6 wt.% sulfur) in dodecane. Dimethyl disulfide was added to the feedstock to maintain the sulfide form of the active component (MoS_2_). The duration of each step was 6 h.

To check catalyst stability, oxygen conversion was compared in the first and last stages carried out in the same conditions in each experiment (290 °C, 3.0 MPa, H_2_/feed ratio of 600 Nm^3^/m^3^, and LHSV of 36 h^−1^).

### 3.5. Product Analysis

The products of methyl palmitate (MP) conversion were analyzed using an Agilent 6890N gas chromatograph (“Agilent Technologies”, USA, Wilmington) equipped with a flame ionization detector and an HP-1MS quartz capillary column (30 m × 0.32 mm × 1 μm). Methyl palmitate conversion was calculated as follows (1):(1)$$X_{MP}=\frac{C_{MP}^{0}-C_{MP}}{C_{MP}^{0}}\times 100\%,$$
where $C_{MP}^{0}$ is the chromatogram peak area of MP in the feed and $C_{O}$ is the chromatogram peak area of MP in the final product.

The total oxygen content in liquid samples was determined using a Vario EL Cube elemental CHNSO analyzer (“Elementar Analysensysteme GmbH”, Langenselbold, Germany).

Oxygen conversion was calculated as (2):(2)$$X_{O}=\frac{C_{O}^{0}-C_{O}}{C_{O}^{0}}\times 100\%,\;\;\;\;$$
where $C_{O}^{0}$ is the total oxygen content in the feed and $C_{O}$ is the total oxygen content in the final product.

The gas phase during the MP hydroprocessing was analyzed online using a gas chromatograph Chromos 1000 (“Chromos”, Omsk, Russia) equipped with a methanator and a flame ionization detector.

The selectivity of the ‘direct’ HDO route (HDO selectivity) was calculated as follows (3):(3)$$S=\frac{C_{16}}{C_{16}+C_{15}}\times 100\%,$$
where *C_16_* is the content of *C_16_* alkanes (normal + iso) in the final product, and *C_15_* is the content of *C_15_* alkanes (normal + iso) in the final product.

The yield of iso-alkanes was calculated as follows (4):(4)$$Y\left(iso\right)=\frac{i-C_{16}+i-C_{15}}{\sum (C_{16}+C_{15)}}*100\%,$$
where *i-C_16_* and i-C_15_ are the contents of *iso*-C_16_ and *iso*-C_15_ alkanes in the final product at complete oxygen conversion; $\sum (C_{16}+C_{15)}$ represents the sum of normal and iso-alkanes at complete oxygen conversion.

## 4. Conclusions

Synthesized composite zeolite-containing supports (30 wt.% zeolite and 70% Al_2_O_3_) and corresponding sulfide Mo-containing catalysts were characterized by XRD, HRTEM, and SEM. According to XRD data, the structure of zeolites was preserved in synthesized supports and catalysts. The uniform distribution of zeolite crystallites in composite materials (Al_2_O_3_-ZSM-5, Al_2_O_3_-ZSM-12, Al_2_O_3_-ZSM-22, and Al_2_O_3_-SAPO-11) was confirmed by SEM-EDX. 100% conversion of oxygen was observed at 310 °C over sulfided Mo/Al_2_O_3_-zeolite catalysts in the hydroprocessing of methyl palmitate. A temperature rise from 310 to 350 °C resulted in a decrease in HDO selectivity. It was found that the addition of zeolite to alumina has a slight influence on MP conversion, but the effect on the conversion of oxygen-containing compounds is greater. The activity of MoS_2_/Al_2_O_3_-zeolite catalysts in the production of isomerized alkanes in MP hydroconversion is in good correlation with the concentration of Brønsted acid sites. The yield of iso-alkanes in the hydroisomerization of MP increased in the following order: Al_2_O_3_ < Al_2_O_3_-ZSM-12 < Al_2_O_3_-ZSM-5 < Al_2_O_3_-SAPO-11 < Al_2_O_3_-ZSM-22. The yield of iso-alkanes was affected by temperature and hydrogen pressure. An increase in temperature and pressure resulted in a decrease in iso-alkane yield. This observation can probably be explained by the reaction mechanism under the given reaction conditions. Iso-alkanes are formed from intermediate products of methyl palmitate HDO, not from alkanes.

## Figures and Tables

**Figure 1 ijms-24-14863-f001:**
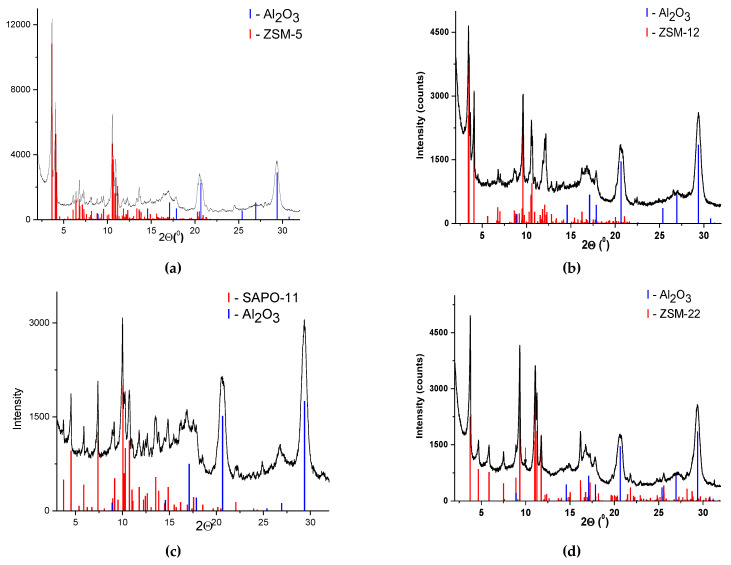
XRD patterns of synthesized Al_2_O_3_-zeolite composite supports ((**a**)—Al_2_O_3_-ZSM-5, (**b**)—Al_2_O_3_-ZSM-12, (**c**)—Al_2_O_3_-SAPO-11, and (**d**)—Al_2_O_3_-ZSM-22).

**Figure 2 ijms-24-14863-f002:**
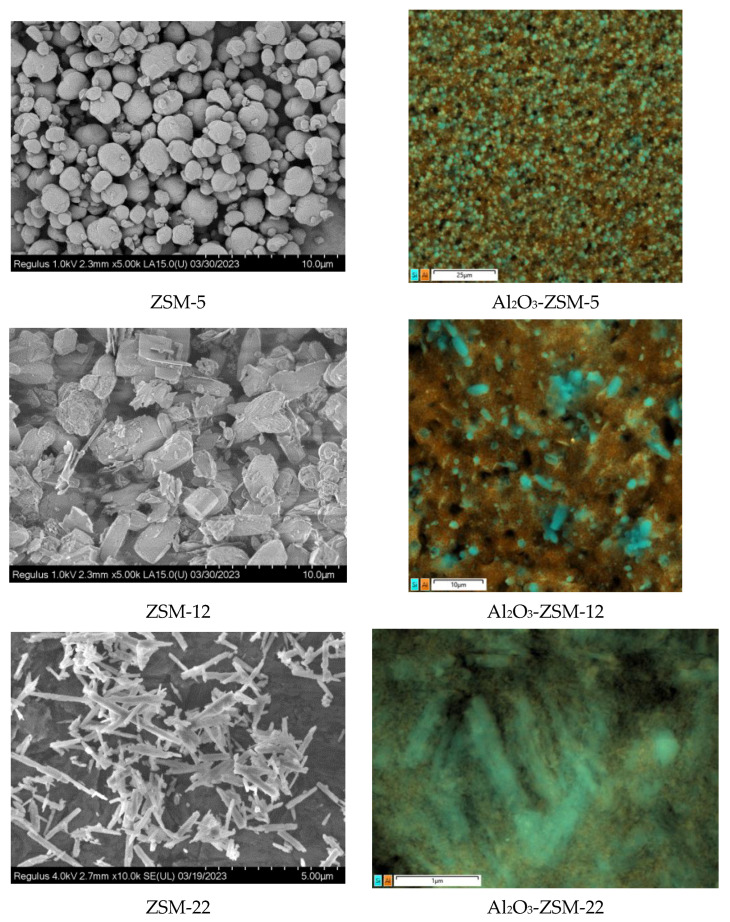
SEM images of zeolite material (**left**) and EDX maps of composite supports of Al_2_O_3_-zeolite (**right**).

**Figure 3 ijms-24-14863-f003:**
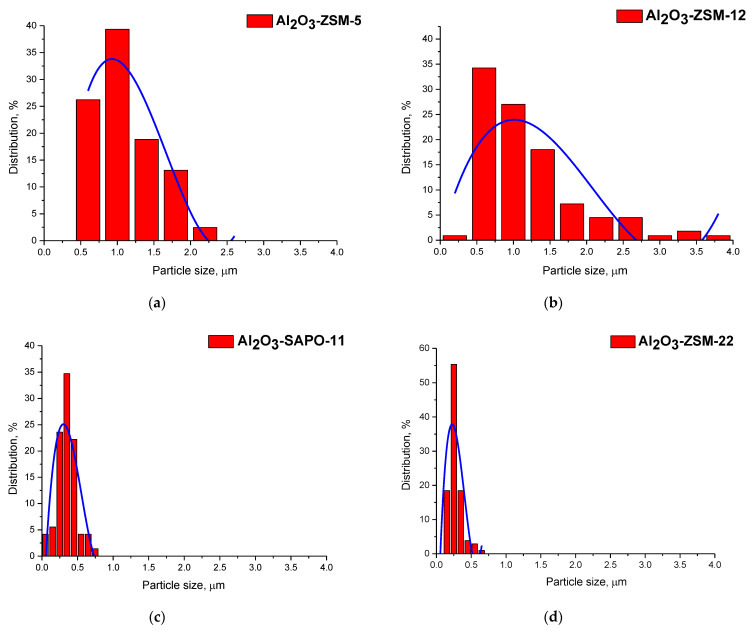
Histograms of zeolite particle size distribution for supports Al_2_O_3_-zeolite (**a**) Al_2_O_3_-ZSM-5 (average particle size of 930 nm), (**b**) Al_2_O_3_-ZSM-12 (average particle size of 1010 nm), (**c**) Al_2_O_3_-SAPO-11 (average particle size of 300 nm), and (**d**) Al_2_O_3_-ZSM-22 (average particle size of 220 nm).

**Figure 4 ijms-24-14863-f004:**
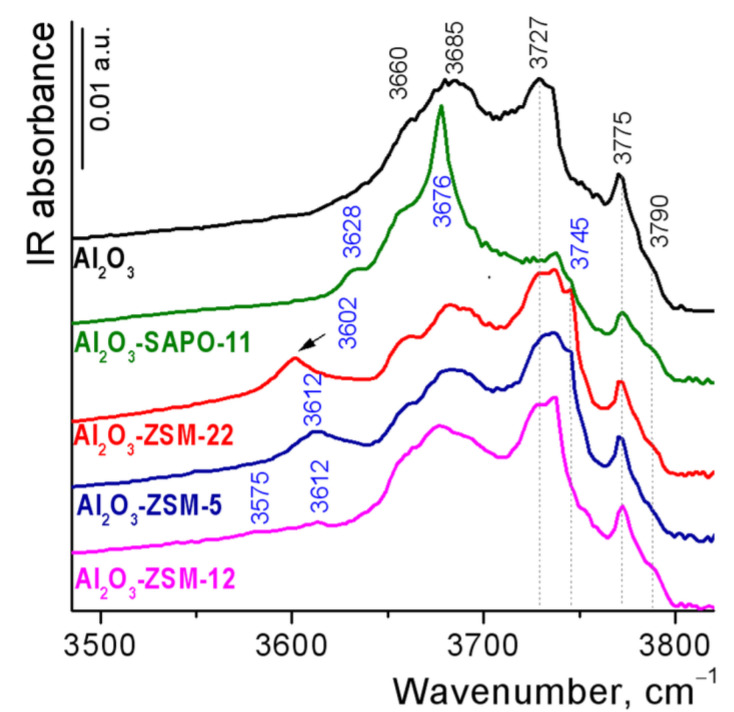
FTIR spectra of Al_2_O_3_ and Al_2_O_3_-zeolite supports after outgassing at 500 °C in the O-H stretching region.

**Figure 5 ijms-24-14863-f005:**
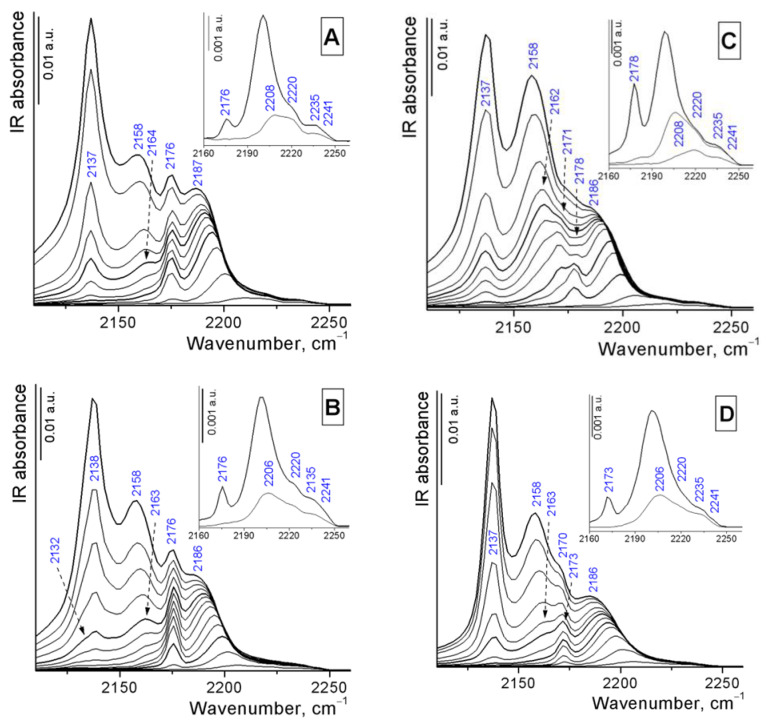
FTIR spectra of CO absorbed at liquid nitrogen temperature on Al_2_O_3_-zeolite supports: Al_2_O_3_-ZSM-5 (**A**), Al_2_O_3_-ZSM-22 (**B**), Al_2_O_3_-ZSM-12 (**C**), and Al_2_O_3_-SAPO-11 (**D**). Equilibrium CO pressures used ranged from 0.1 (bottom curve) to 5 mbar (top curve). The inset shows enlarged spectra at low CO pressure. All spectra are background-corrected.

**Figure 6 ijms-24-14863-f006:**
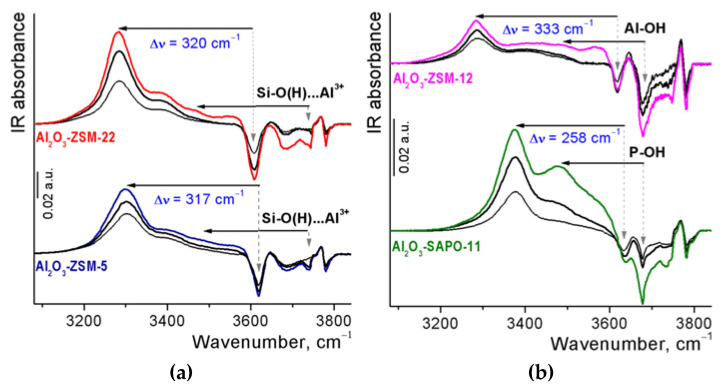
FTIR difference spectra of the OH stretching region during adsorption of CO at liquid nitrogen temperature and low equilibrium CO pressure of 0.3, 0.5, and 1 mbar on Al_2_O_3_-zeolite supports: (**a**) Al_2_O_3_-ZSM-22 and Al_2_O_3_-ZSM-5; (**b**) Al_2_O_3_-ZSM-12 and Al_2_O_3_-SAPO-11.

**Figure 7 ijms-24-14863-f007:**
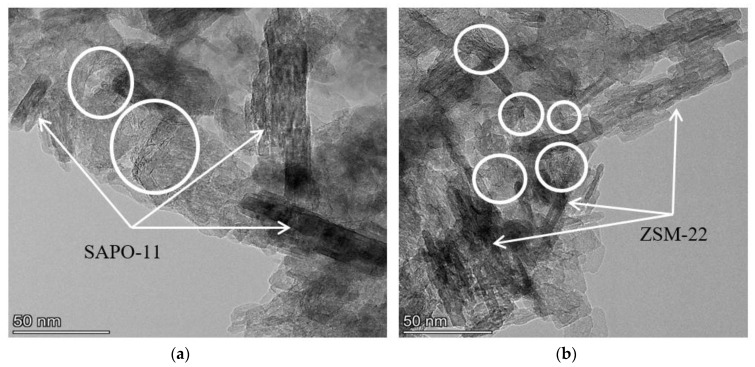
HRTEM images of MoS_2_/Al_2_O_3_-SAPO-11 (**a**) and MoS_2_/Al_2_O_3_-ZSM-22 (**b**) catalysts (circle—sulfide nanoparticles on the alumina surface).

**Figure 8 ijms-24-14863-f008:**
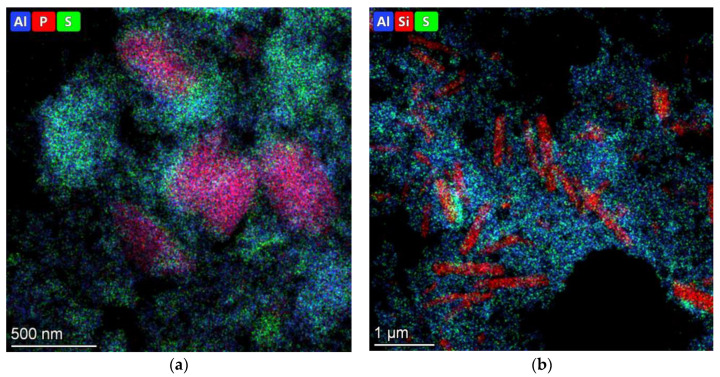
HRTEM-EDX mapping of MoS_2_/Al_2_O_3_-SAPO-11(**a**) and MoS_2_/Al_2_O_3_-ZSM-22 (**b**) catalysts.

**Figure 9 ijms-24-14863-f009:**
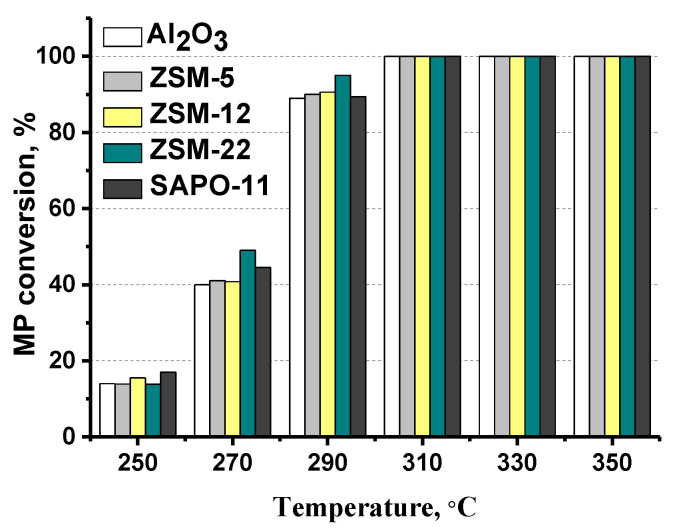
MP conversion in dependence on temperature over MoS_2_/Al_2_O_3_-zeolite catalysts (reaction conditions: 250–350 °C, 3.0 MPa, 600 Nm^3^/m^3^, 36 h^−1^).

**Figure 10 ijms-24-14863-f010:**
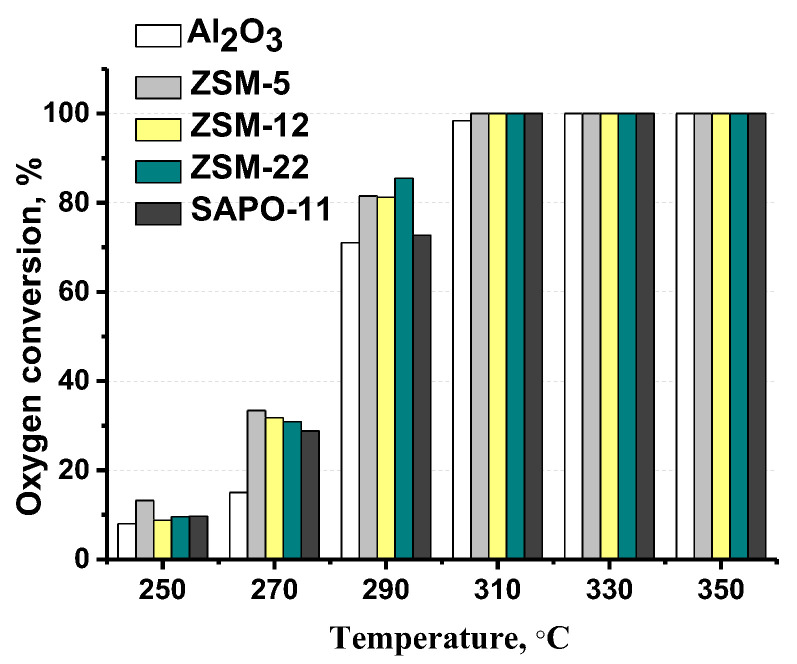
The conversion of oxygen-containing compounds in dependence on temperature over MoS_2_/Al_2_O_3_-zeolite catalysts (reaction conditions: 250–350 °C, 3.0 MPa, 600 Nm^3^/m^3^, 36 h^−1^).

**Figure 11 ijms-24-14863-f011:**
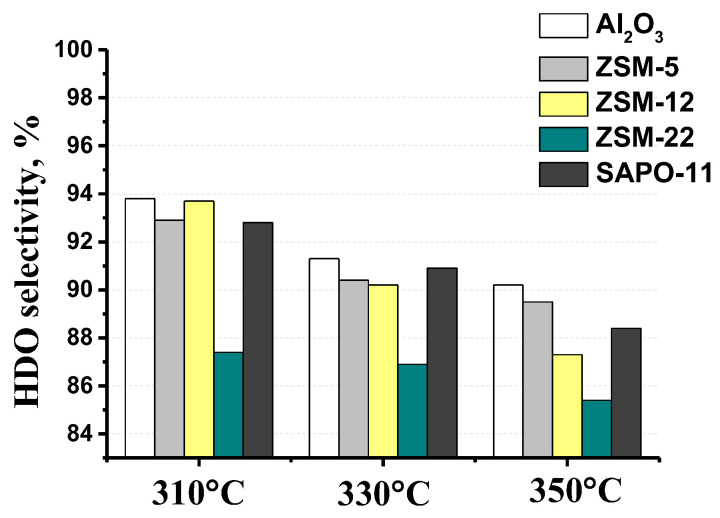
Temperature effect on HDO selectivity over MoS_2_/Al_2_O_3_-zeolite catalysts (reaction conditions: 310–350 °C, 3.0 MPa, 600 Nm^3^/m^3^, 36 h^−1^).

**Figure 12 ijms-24-14863-f012:**
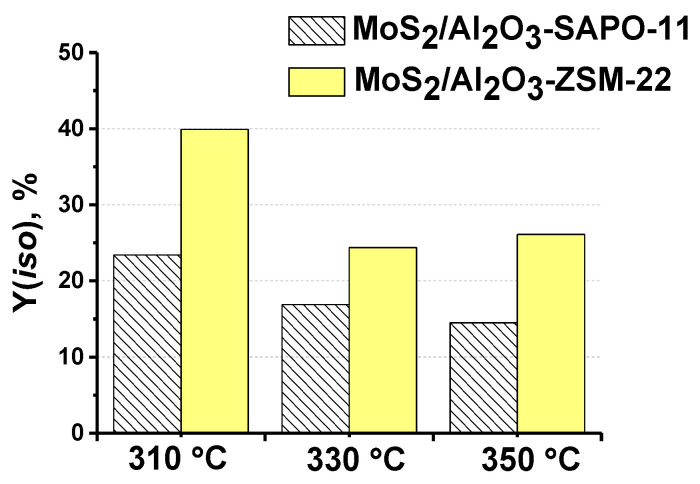
Temperature effect on yield of isomeric C_16_H_34_ and C_15_H_32_ alkanes over MoS_2_/Al_2_O_3_-SAPO-11 and MoS_2_/Al_2_O_3_-ZSM-22 catalysts (reaction conditions: 310–350 °C, 3.0 MPa, 600 Nm^3^/m^3^, 36 h^−1^).

**Figure 13 ijms-24-14863-f013:**
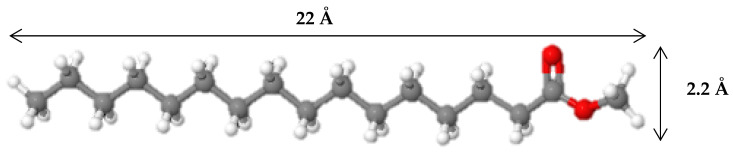
Dimensions of methyl palmitate. Gray, white, and red balls are carbon, hydrogen, and oxygen atoms, respectively [55,56].

**Figure 14 ijms-24-14863-f014:**
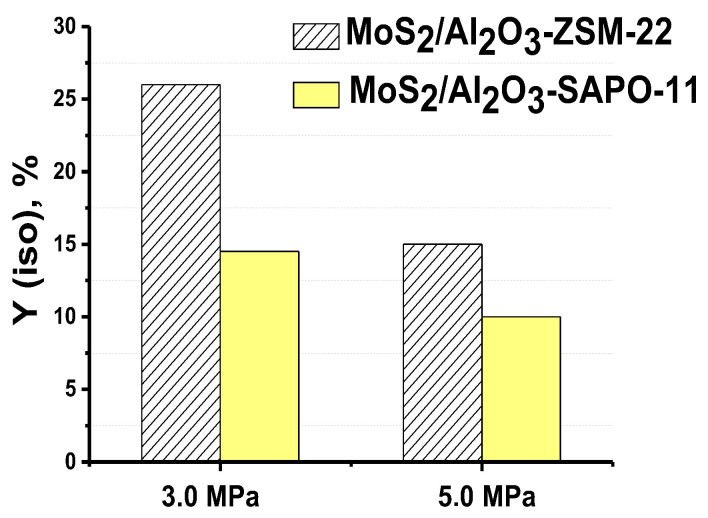
Pressure effect on yield of isomeric C_16_H_34_ and C_15_H_32_ alkanes over MoS_2_/Al_2_O_3_-SAPO-11 and MoS_2_/Al_2_O_3_-ZSM-22 catalysts (reaction conditions: 350 °C, 3.0 and 5.0 MPa, 600 Nm^3^/m^3^, 36 h^−1^).

**Table 1 ijms-24-14863-t001:** The properties of the prepared catalysts.

Catalysts	Mo, wt.%	Support	Textural Properties of the Support		
			Surface Area, m^2^/g	Pore Volume, cm^3^/g	Pore Diameter, nm
Mo/Al_2_O_3_	6.95	Al_2_O_3_	133	0.66	25.1
Mo/Al_2_O_3_-ZSM-5	6.90	Al_2_O_3_-ZSM-5	202	0.48	25.6
Mo/Al_2_O_3_-ZSM-12	6.96	Al_2_O_3_-ZSM-12	165	0.49	22.8
Mo/Al_2_O_3_-ZSM-22	6.90	Al_2_O_3_-ZSM-22	175	0.53	25.5
Mo/Al_2_O_3_-SAPO-11	6.97	Al_2_O_3_-SAPO-11	177	0.42	22.6

**Table 2 ijms-24-14863-t002:** Type, acid strength, and concentration of Brønsted acid sites of Al_2_O_3_-zeolite composite supports.

Al_2_O_3_-Zeolite Composites	Type of Zeolite Sites	IR Frequency Shift/cm^−1^		BAS Concentration (μmol g^−1^)
		Δν_OH…CO_ *^a^*	Δν_CO_ *^b^*	
Al_2_O_3_-ZSM-5	Framework Si-O(H)-Al groups	–317		8.1
	Extra-framework Si-O(H)…Al^3+^ groups	–(260 ÷ 270)	+33	1.8
Al_2_O_3_-ZSM-22	Framework Si-O(H)-Al groups	–320		11.4
	Extra-framework Si-O(H)…Al^3+^ groups	–(260 ÷ 270)	+33	1.0
Al_2_O_3_-ZSM-12	Framework Si-O(H)-Al groups	–333	+35	2.7
	Extra-framework Al-OH groups	–(200÷196)	+28	2.6
Al_2_O_3_-SAPO-11	Framework Si-O(H)-Al groups	–258	+30	7.4
	P-OH groups	–(202÷198)	+27	~4 *^c^*

*^a^* Red frequency shift of the bands of O-H groups at hydrogen bonding with CO. *^b^* Blue frequency shift of the CO stretching bands at hydrogen bonding of CO with OH- groups relative to the gas phase CO. *^c^* The contribution of the P-OH groups for the SAPO-11 is approximate (molar integral absorption coefficients for the P-OH group are unknown).

**Table 3 ijms-24-14863-t003:** List of zeolites relevant to this work with details.

Material	SiO_2_/Al_2_O_3_ Mole Ratio	Framework Type	Channels	Size of Channels
ZSM-5	280	MFI	3D, 10 MR	5.3 × 5.6 Å [010] 5.1 × 5.5 Å [100] [58]
ZSM-12	280	MTW	1D, 12 MR	5.6 × 7.7 Å [010] [59]
ZSM-22	97	TON	1D, 10 MR	4.6 × 5.7 Å [001] [58]
SAPO-11	SiO_2_/Al_2_O_3_/P_2_O_5_ = 0.25/1.0/0.8	AEL	1D, 10 MR	3.9 × 6.3 Å [001] [60]

## Data Availability

Data is available upon request from the corresponding authors.

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
