# Peer review of "Bifunctional MoS2/Al2O3-Zeolite Catalysts in the Hydroprocessing of Methyl Palmitate"

_ijms, 2023, doi:10.3390/ijms241914863_

Round 1
Reviewer 1 Report
Overall, this paper presents an interesting study on the synthesis and characterization of a series of bifunctional catalysts for the hydroconversion of methyl palmitate. The use of MoS2/Al2O3-zeolite catalysts with different zeolite supports (ZSM-5, ZSM-12, ZSM-22, and SAPO-11) is well-presented, and the results demonstrate promising catalytic performance in terms of hydrodeoxygenation (HDO) and isomerization activities. The effects of temperature and pressure were also well discussed.
Therefore, I support its publication in this journal without further modifications.
English is fine
Author Response
Thank you very much for taking the time to review our manuscript!
Reviewer 2 Report
The manuscript "Bifunctional MoS2/Al2O3-zeolite catalysts in hydroprocessing of methyl palmitate ", in the reviewer opinion, has high potential to inform readers about future technology of possible manufacturing of diesel like bio-fuels via hydrotreatment of fatty acid esters over MoS2-based catalysts. The manuscript covers topical aspects of the shift of the petroleum-based refinery to modern and sustainable biorefinery. The manuscript could have been published in the present form. However, there are a few issues that could be addressed by the authors within minor revisions that should clarify the scientific content:
(1) L123: “aliphatic esters “ instead of “aliphatic ethers”?
(2) Introduction, or elsewhere in the manuscript: High isomerization activity is a function of g Brønsted acid sites (BAS) or simply acidity of the most Si-Al oxides. Previous papers solved similar topic using acidic silica MCM-41 and supported Mo, CoMo and NiMo sulfides (Reaction Kinetics, Mechanisms and Catalysis (2019) 127:887–902). Octanoic acid, as a model compound of HDO reaction, formed the predominant amount of branched hydrocarbons (both olefins and iso-alkanes) using these catalysts. Furthermore, cracking was also observed despite the fact that MCM-41 is often referred being without (or without considerable) Brønsted acidity. Nevertheless, the acidity of MCM-41 was obviously of “zeolite-like” acidity because MCM-41 cracked also cumene in this paper. The behavior of forming branched hydrocarbons by HDO was not observed for other supports such as MgO, Al2O3, ZrO2,TiO2. The authors may consider to enrich the content of their manuscript using this reference. Moreover, HDO of the studied longer fatty acid ester point on the advantages of using moderate acidity catalysts to crack and to isomerate to make final diesel of more liquid, i.e. without extended amount of too long linear paraffins.
(3) Experimental/the catalytic test: It is not clear whether a source of sulfur-containing compounds was used during the HDO to keep the MoS2 catalysts sulphidic. Did the author observed any oxidation of MoS2 surface during the HDO if a source of sulphur were not used?
To sum up, minor revisions were suggested because no additional experimental work is need, in the reviewer opinion.
Author Response
Thank you very much for taking the time to review our manuscript.
Response 1: L123: “aliphatic esters “ instead of “aliphatic ethers”?
Thank you for pointing this out. We agree with this comment. We've replaced 'ethers' with 'esters' throughout the text of the manuscript.
Response 2: Introduction, or elsewhere in the manuscript: High isomerization activity is a function of g Brønsted acid sites (BAS) or simply acidity of the most Si-Al oxides. Previous papers solved similar topic using acidic silica MCM-41 and supported Mo, CoMo and NiMo sulfides (Reaction Kinetics, Mechanisms and Catalysis (2019) 127:887–902). Octanoic acid, as a model compound of HDO reaction, formed the predominant amount of branched hydrocarbons (both olefins and iso-alkanes) using these catalysts. Furthermore, cracking was also observed despite the fact that MCM-41 is often referred being without (or without considerable) Brønsted acidity. Nevertheless, the acidity of MCM-41 was obviously of “zeolite-like” acidity because MCM-41 cracked also cumene in this paper. The behavior of forming branched hydrocarbons by HDO was not observed for other supports such as MgO, Al2O3, ZrO2,TiO2. The authors may consider to enrich the content of their manuscript using this reference. Moreover, HDO of the studied longer fatty acid ester point on the advantages of using moderate acidity catalysts to crack and to isomerate to make final diesel of more liquid, i.e. without extended amount of too long linear paraffins.
Thank you for pointing this out. We agree with your comment. We've added this reference in our manuscript in section 'Introduction'. Corrections were highlighted in the re-submitted manuscript.
Response 3: Experimental/the catalytic test: It is not clear whether a source of sulfur-containing compounds was used during the HDO to keep the MoS2 catalysts sulphidic. Did the author observed any oxidation of MoS2 surface during the HDO if a source of sulphur were not used?
Thank you for pointing this out. We did not indicate in section '2.4. Catalytic experiments' that dimethyl disulfide was added to preserve sulfide form of active component. Now we've corrected it. Corrections were highlighted in the re-submitted manuscript.